# Conformalized Fairness via Quantile Regression

**Meichen Liu**[1], **Lei Ding**[1], **Dengdeng Yu**[2], **Wulong Liu**[3],
**Linglong Kong**[1]*, **Bei Jiang**[1]*
[1]Department of Mathematical and Statistical Sciences, University of Alberta
[2]Department of Mathematics, University of Texas at Arlington
[3] Huawei Noah's Ark Lab Canada
{meichen1,lding1,lkong,bei1}@ualberta.ca
{dengdeng.yu}@uta.edu
{liuwulong}@huawei.com

## Abstract

Algorithmic fairness has received increased attention in socially sensitive domains. While rich literature on mean fairness has been established, research on quantile fairness remains sparse but vital. To fulfill great needs and advocate the significance of quantile fairness, we propose a novel framework to learn a real-valued quantile function under the fairness requirement of *Demographic Parity* with respect to sensitive attributes, such as race or gender, and thereby derive a reliable *fair* prediction interval. Using optimal transport and functional synchronization techniques, we establish theoretical guarantees of distribution-free coverage and exact fairness for the induced prediction interval constructed by fair quantiles. A hands-on pipeline is provided to incorporate flexible quantile regressions with an efficient fairness adjustment post-processing algorithm. We demonstrate the superior empirical performance of this approach on several benchmark datasets. Our results show the model's ability to uncover the mechanism underlying the fairness-accuracy trade-off in a wide range of societal and medical applications.

## 1   Introduction

We are increasingly leaning on machine learning systems to tackle human problems. A primary objective is to develop intelligent algorithms that can automatically produce accurate decisions which also enjoy equitable properties, as unintended social bias has been identified as a rising concern in various fields [13, 18, 14].

As a means of providing quantitative measures of fairness, a number of metrics have been proposed. These metrics can be categorized into three broad categories: group fairness [3], individual fairness [25], and causality-based fairness [31]. In contrast to causality-based fairness that requires domain knowledge to develop a fair causal structure and individual fairness that seeks equality only between similar individuals, group fairness does not require any prior knowledge and seeks equality for groups as a whole [6]. Among the metrics defined for group fairness such as equalized odds [9, 29] and predictive rate parity [10], demographic parity (DP) is generic since it does not allow prediction results in aggregate to depend on sensitive attributes [1, 21, 12, 39]. In particular, an algorithm is said to satisfy DP if its prediction is independent of any given sensitive attribute [1].

There have been a number of studies on algorithmic fairness concerning DP [1, 11, 12, 21, 32, 39]. In the context regression analysis, much attention have been paid on conditional mean inferences [1, 11, 12, 32], few are concerned with conditional quantiles [37, 39]. As real-world data often exhibit heterogeneity, contain extreme outliers, or do not meet satisfactory distributional assumptions,

---

*Co-Corresponding Authors

like Gaussianity, a fairness discussion on conditional quantiles may be more rational and essential since they are able to provide a more complete understanding of the dependence structure between response and explanatory variables [39], as well as better accommodate asymmetry and extreme tail behavior [38]. It should also be noted that bias or unfairness that arises in mean regression may also be propagated through quantile regression, therefore it must be properly dealt with separately: a graphic demonstration can be found in Figure 1. More intuitively, we may take an example from a Spanish labor market study [17, 18]. The study found that in Spain, also in line with other countries, the mean wage gap between men and women is quite substantial: on average, women earn around 70 percent of what men earn. While wage gaps are not uniform across all pay scales, they are greater at higher quantiles than at lower quantiles. As biases and disparities at different quantiles tend to be overshadowed by the mean behavior of the entire population, we propose a novel framework to seek fair predictions at different quantiles. It uses optimal transport techniques [2, 12] by transforming *bias-affected* distributions into an *only-fair* Wasserstein-2 barycenter through a kernel-based functional synchronization method [8, 45], in order to provide fair quantile estimators.

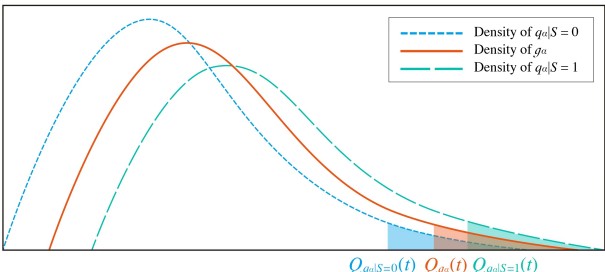

Figure 1: An illustration of quantile fairness: for a skewed and heteroscedastic quantile estimation $\{q_{\alpha,i}\}_{i=1}^{N}$ affected by the sensitive attribute $S \in \{0, 1\}$, for example, the higher quantile of the salary distribution, the optimal fair quantile prediction $Q_{g_\alpha}(t), t \in (0, 1)$ is derived through a convex combination of the conditional quantile functions of $Q_{q_\alpha|S=0}$ and $Q_{q_\alpha|S=1}$.

Since quantile fairness poses a number of theoretical challenges, no previous literature has been able to provide any inference results such as prediction intervals concerning quantile fairness. It is imperative to keep in mind that fairness is only one of two legs of the primary goal of modern machine learning algorithms, the other being accuracy. Building a reliable prediction with valid confidence is a significant challenge that is encountered by many machine learning algorithms [44]. Towards this end, we propose the conformalized fair quantile prediction (CFQP) inspired by the works of Romano et al. [33, 34]. Our analysis demonstrates, both mathematically and experimentally, that CFQR provides finite sample, distribution-free validity, DP fairness for different quantiles, and precise control of the miscoverage rate, regardless of the underlying quantile algorithm.

**Contributions and Outlines.** In this paper, we propose a new quantile based method with valid inference that enhances both accuracy and fairness while maintaining a balance between the two. It is a novel framework that allows an exact control of prediction miscoverages while ensuring quantile fairness simultaneously. The main contributions are summarized as follows:

i. We successfully transform the problem of searching quantiles under DP fairness to the construction of multi-marginal Wasserstein-2 barycenters via the optimal transport theory [2, 12, 19]. We incorporate a novel kernel smoothing step into the preceding method, which is particularly advantageous for subgroups whose sample sizes are too small to obtain reliable quantile function estimations.

ii. In Section 4, we propose a conformalized fair quantile regression prediction interval (CFQP) inspired by the works of Romano et al. [33, 34]. It is mathematically proved to achieve a distribution-free validity, demographic parity on different quantiles, and an exact control of miscoverage rates, regardless of the quantile algorithm used. The theoretical validity of prediction interval constructed by CFQP and exact DP of the fair quantile estimators are given in Section 5 and the supplement.

iii. The experimental results presented in Section 6 include a numerical comparison of the proposed CFQP and fair quantile estimation with both state-of-the-art conformal and fairness-oriented

methods. By reducing the discriminatory bias dramatically, our method outperforms the state-of-the-art methods while maintaining reasonable short interval lengths.

**Related works.** Existing approaches for building a fair mean regression broadly fall into three classes: pre-processing, in-processing and post-processing. In particular, preprocessing methods focus on transforming the data to remove any unwanted bias [5, 31, 43]; in-processing methods aim to build in fairness constraints into the training step [1, 4, 25, 29]; post-processing methods target to modify the trained predictor [11, 12, 28]. As few previous works have focused on the quantile fairness of and fair prediction interval, the most related are Yang et al. [39], where a different fairness measure was used. While Agarwal et al. [1] mentioned that their reduction-based approach can be adapted into quantile regression, Williamson and Menon [37] brought forward a novel conditional variance at risk fairness measure aiming to control the largest subgroup risk. For interval fairness measure, the approach by Romano et al. [33] achieved equalized coverage among groups without fairness on interval endpoints. Methodologically, integrating algorithmic fairness with Wasserstein distance based barycenter problem has been studied in [2, 11, 12, 19, 23]. Both in-processing [1, 23] and post-processing [11, 12] methods were proposed to solve classification and mean regression problems. As a post-processing method, our work is distinct from above-mentioned methods by constructing the DP-fairness for each population quantile, and generating a fair prediction interval accordingly.

**Notations.** We denote by $[K]$ the set $\{1, \ldots, K\}$ for arbitrary integer $K$. $|\mathcal{S}|$ represents the cardinality for a finite set $\mathcal{S}$. $E$ and $P$ represent the expectation and probability and $\mathbb{1}\{\cdot\}$ is the indicator function. Let $\{Z_n\}_{n=1}^{\infty}$ be a sequence of random variables, and $\{k_n\}_{n=1}^{\infty}$ be a sequence of positive numbers, we say that $Z_n = O_p(k_n)$, if $\lim_{T \to \infty} \lim \sup_{n \to \infty} P(|Z_n| > Tk_n) = 0$, then $Z_n/k_n = O_p(1)$. To denote the equality in distribution of two random variables $A$ and $B$, we write $A \overset{d}{=} B$.

## 2   Problem statement

Consider the regression problem where a "sensitive characteristic" $S$ is available, which by the U.S. law [19, 33] can be enumerated as sex, race, age, disability, etc. We observe the triplets $(X_1, S_1, Y_1), \ldots, (X_n, S_n, Y_n)$, denote $(X_i, S_i, Y_i)$ by $Z_i$, $i = 1, \ldots, n$ and $Z_i$ is a random variable in $\mathbb{R}^p \times [K] \times \mathbb{R}$. The aim is to predict the unknown value of $Y_{n+1}$ at a test point $X_{n+1}, S_{n+1}$. Let $P$ be the joint distribution of $Z$, we assume that all the samples $\{Z_i\}_{i=1}^{n+1}$ are drawn exchangeable, where i.i.d. is a special case.

Our goal is to construct a marginal distribution-free prediction band $C(X_{n+1}, S_{n+1}) \subseteq \mathbb{R}$ that is likely to cover the unknown response $Y_{n+1}$ with finite-sample (nonasymptotic) validity. Formally, given a desired miscoverage rate $\alpha$, the predicted interval satisfies

$$P\{Y_{n+1} \in C(X_{n+1}, S_{n+1})\} \geq 1 - \alpha \tag{1}$$

for any joint distribution $P$ and any sample size $n$, while the left and right endpoint of $C(X_{n+1}, S_{n+1})$ satisfies the fairness constraint of Demographic Parity concerning the sensitive variable $S$.

**Demographic Parity.** We introduce the quantitative definition of DP in fair regression and connect the DP-fairness with a quantile regressor $q_\alpha$. The result that $q_\alpha$ can be projected to the fair counterparts using optimal transport will be invoked later.

Given a fixed quantile level $\alpha$ (it may refer to $\alpha_{\text{lo}}$ or $\alpha_{\text{hi}}$ indicating the upper and lower quantile estimates for the prediction band $C(X_{n+1}, S_{n+1})$). Let $q_\alpha : \mathbb{R}^p \times [K] \to \mathbb{R}$ represent an arbitrary conditional quantile predictor. Denote by $\nu_{q_\alpha|s}$ the distribution of $(q_\alpha(X, S) \mid S = s)$, the Cumulative Distribution Function (CDF) of $\nu_{q_\alpha|s}$ is given by

$$F_{\nu_{q_\alpha|s}}(t) = P(q_\alpha(X, S) \leq t \mid S = s). \tag{2}$$

The quantile function $Q_{\nu_{q_\alpha|s}} = F_{\nu_{q_\alpha|s}}^{-1} : [0, 1] \to \mathbb{R}$, namely, the generalized inverse of $F_{\nu_{q_\alpha|s}}$, can thus be defined as for all levels $t \in (0, 1]$,

$$Q_{\nu_{q_\alpha|s}}(t) = \inf\{y \in \mathbb{R} : F_{\nu_{q_\alpha|s}}(y) \geq t\} \text{ with } Q_{\nu_{q_\alpha|s}}(0) = Q_{\nu_{q_\alpha|s}}(0+). \tag{3}$$

To simplify the notations, we will write $F_{q_\alpha|s}$ and $Q_{q_\alpha|s}$ instead of $F_{\nu_{q_\alpha|s}}$ and $Q_{\nu_{q_\alpha|s}}$ respectively, for any prediction rule $q_\alpha$.

In the following, we introduce the definition of Demographic Parity (DP), which is most commonly used in the context of fairness research [1, 11, 12, 21, 29].

**Definition 1** (Demographic Parity). An arbitrary prediction $g : \mathbb{R}^d \times [K] \to \mathbb{R}$ satisfies demographic parity under a distribution $P$ over $(X, S, Y)$, if $g(X, S)$ is statistically independent of the sensitive attribute $S$. Formally, for every $s, s' \in [K]$,

$$\sup_{t \in \mathbb{R}} |P(g(X, S) \leq t \mid S = s) - P(g(X, S) \leq t \mid S = s')| = 0.$$

Demographic Parity (DP) requires the predictions to be independent of the sensitive attribute, and it demands the Kolmogorov-Smirnov distance [26] (the difference between CDFs measured in the $l_\infty$ norm) between $\nu_{g|s}$ and $\nu_{g|s'}$ to vanish for all categories $s, s'$.

## 3 Quantile Regression and Conformal Prediction

In this section, we recall the CQR approach for finite sample, distribution-free prediction interval inference. Quantile regression was proposed by Koenker and Bassett [24] to estimate the $\alpha$-th quantile of the conditional distribution of $Y$ given $\tilde{X} := (X, S)$ for some quantile level $\alpha \in (0, 1)$, since then it has become more pervasive with various applications, such as providing prediction intervals, detecting outliers, or perceiving the entire distribution [27, 20]. Denote the conditional cumulative distribution of $Y$ given $\tilde{X}$ by $F(y \mid \tilde{X} = \tilde{x}) := P\{Y \leq y \mid \tilde{X} = \tilde{x}\}$. The $\alpha$-th conditional quantile prediction is defined as $q_\alpha(\tilde{x}) := \inf\{y \in \mathbb{R} : F(y \mid \tilde{X} = \tilde{x}) \geq \alpha\}$. Quantile regression can be cast as an optimization problem [27, 42, 30, 36, 41], by minimizing the expected check loss function $E(\rho_\alpha) = E[\rho_\alpha(y, q) | \tilde{X} = \tilde{x}]$, where

$$\rho_\alpha(y, q_\alpha(\tilde{x})) = \begin{cases} \alpha|y - q_\alpha(\tilde{x})| & \text{if } y \geq q_\alpha(\tilde{x}), \\ (1 - \alpha)|y - q_\alpha(\tilde{x})| & \text{if } y < q_\alpha(\tilde{x}). \end{cases} \tag{4}$$

Quantile regression offers a principled way of judging the reliability of predictions by building a prediction interval for the new observation $(\tilde{X}_{n+1}, Y_{n+1})$. In contrast to asymptopia, Romano et al. [33, 34] brought forward the conformalized quantile regression (CQR) by combining the merits of robust quantile regression with conformal prediction, thus finite sample validity in Eq. (1) is guaranteed. Inspired by the split conformal method, a split CQR likewise starts with splitting the data into a proper training set and a calibration set, indexed by $\mathcal{I}_1, \mathcal{I}_2$ respectively. Given any quantile regression algorithm $\mathcal{Q}$, we then fit two conditional quantile functions $\hat{q}_{\alpha_{\text{lo}}}$ and $\hat{q}_{\alpha_{\text{hi}}}$ on the proper training set: $\{\hat{q}_{\alpha_{\text{lo}}}, \hat{q}_{\alpha_{\text{hi}}}\} \leftarrow \mathcal{Q}\left(\left\{\left(\tilde{X}_i, Y_i\right) : i \in \mathcal{I}_1\right\}\right)$. The conformity scores are calculated to quantify the error made by the plug-in prediction interval $\hat{C}(\tilde{x}) = [\hat{q}_{\alpha_{\text{lo}}}(\tilde{x}), \hat{q}_{\alpha_{\text{hi}}}(\tilde{x})]$. We evaluate the scores on the calibration set as $E_k := \max\left\{\hat{q}_{\alpha_{\text{lo}}}(\tilde{X}_k) - Y_k, Y_k - \hat{q}_{\alpha_{\text{hi}}}(\tilde{X}_k)\right\}$ for each $k \in \mathcal{I}_2$, where both undercoverage and overcoverage of the interval are taken into consideration [34]. Given a new input data $\tilde{X}_{n+1}$, we construct the prediction interval for $Y_{n+1}$ as $C\left(\tilde{X}_{n+1}\right) = \left[\hat{q}_{\alpha_{\text{lo}}}\left(\tilde{X}_{n+1}\right) - Q_{1-\alpha}\left(E, \mathcal{I}_2\right), \hat{q}_{\alpha_{\text{hi}}}\left(\tilde{X}_{n+1}\right) + Q_{1-\alpha}\left(E, \mathcal{I}_2\right)\right]$, where $Q_{1-\alpha}(E, \mathcal{I}_2) := (1 - \alpha)(1 + 1/|\mathcal{I}_2|)$-th empirical quantile of $\{E_k : k \in \mathcal{I}_2\}$ conformalizes the plug-in prediction interval. Note that the constructed interval $C(\tilde{X}_{n+1})$ could be highly influenced by the sensitive variable $S$.

## 4 Conformal fair quantile prediction (CFQP)

We formally describe our proposed conformal fair prediction (CFQP) framework for constructing DP fairness constrained prediction intervals in this section. A kernel smoothing quantile function is introduced during the functional synchronization, which can improve the estimation when some subgroups are too small to give reliable sample quantile function estimations.

**Definition 2** (Wasserstein-2 distance). Let $\mu$ and $\nu$ be two univariate probability measures with finite second moments. The squared Wasserstein-2 distance between $\mu$ and $\nu$ is defined as

$$\mathcal{W}_2^2(\mu, \nu) = \inf\left\{\int_{\mathbb{R} \times \mathbb{R}} |x - y|^2 d\gamma(x, y), \gamma \in \Gamma_{\mu, \nu}\right\}$$

where $\Gamma_{\mu, \nu}$ is the set of probability measures (couplings) on $\mathbb{R} \times \mathbb{R}$ having $\mu$ and $\nu$ as marginals.

**Proposition 1** (Fair optimal prediction [12])**.** Assume, for each $s \in [K]$, that the univariate measure $\nu_{q_\alpha|s}$ has a density and let $p_s = P(S = s)$. Then,

$$\min_{g_\alpha \text{ is fair}} E\left(q_\alpha(X, S) - g_\alpha(X, S)\right)^2 = \min_\nu \sum_{s \in [K]} p_s \mathcal{W}_2^2\left(\nu_{q_\alpha|s}, \nu\right). \tag{5}$$

Moreover, if $g_\alpha$ and $\nu$ solve the l.h.s. and the r.h.s. problems respectively, then $\nu = \nu_{g_\alpha}$ and specifically,

$$g_\alpha(x, s) = \sum_{s' \in [K]} p_{s'} Q_{q_\alpha|s'} \circ F_{q_\alpha|s} \circ q_\alpha(x, s). \tag{6}$$

Proposition 1 implies that the optimal fair quantile predictor for an input $(x, s)$ is obtained by a non-linear transformation of the vector $[q_\alpha(x, s)]_{s=1}^K$ linking to a Wasserstein barycenter problem[2, 12]. The explicit closed form solution comes from [2, 12, 16], which relies on the classical characterization of optimal coupling in one dimension of the Wasserstein-2 distance. A rigorous proof is given in [12, 23]. It shows that a minimizer $g_\alpha$ of the $L_2$-risk can be used to construct $\nu$ and vice-versa, given $\nu$, there is a explicit expression Eq. (6) for the multi-marginal Wasserstein barycenter [2].

First, we provide a sketch of the CFQP approach. We start with splitting the whole training data into a proper training set $\mathcal{I}_1$ and a calibration set $\mathcal{I}_2$, then fit an arbitrary quantile regression algorithm $\mathcal{Q}$ on $\mathcal{I}_1$, $\{\hat{q}_{\alpha_{\text{lo}}}, \hat{q}_{\alpha_{\text{hi}}}\} \leftarrow \mathcal{Q}\left(\left\{\left(\tilde{X}_i, Y_i\right) : i \in \mathcal{I}_1\right\}\right)$. We apply the fitted quantile algorithm $\mathcal{Q}$ on the calibration set $\mathcal{I}_2$ to obtain the predicted $\{\hat{q}_{\alpha_{\text{lo}}}(\tilde{X}_i), \hat{q}_{\alpha_{\text{hi}}}(\tilde{X}_i)\}_{i \in \mathcal{I}_2}$. Since the quantile estimates for $\mathcal{I}_2$ will be used for conformalization, it is essential to transform them into fair ones, i.e. $\hat{g}_{\alpha,i}, \forall i \in \mathcal{I}_2$ (Eq. (9)), through Algorithm 2. Finally, for a test point $\tilde{X}_{n+1}$, we will predict two quantile estimates $\hat{q}_\alpha(x, s)$ affected by the sensitive variable $S$ by $\mathcal{Q}$, then apply the functional synchronization (details in Algorithm 2) and calibration (Algorithm 1) steps in turn to generate the fair constraint prediction interval $C(\tilde{X}_{n+1})$ for $Y_{n+1}$.

Next, we explicate in detail how to remove the effect of the sensitive variable for the predicted quantile estimates. By Proposition 1, the optimal fair quantiles take the form of Eq. (6). Therefore, we propose an empirical optimal fair quantile estimator $\hat{g}_\alpha$ that relies on the plug-in principle. In particular, Eq.(6) indicates that for each quantile level $\alpha$ and each category $s \in [K]$, we only need estimators for the regression function $\hat{q}_\alpha$, the proportion $\hat{p}_s$, the cumulative distribution function $F_{\hat{q}_\alpha|s}$ and the quantile function $Q_{\hat{q}_\alpha|s}$.

Note that we can empirically estimate the CDF and quantile function for each sensitive group in the calibration set $\mathcal{I}_2$ separately. Hence for each quantile level $\alpha$, let $N_s := |\mathcal{I}_2^s|$, and the quantile estimators $(\hat{q}_1^s, \hat{q}_2^s, \ldots, \hat{q}_{N_s}^s)^2$ are calculated through the fitted quantile regression $\mathcal{Q}$ with training set $\mathcal{I}_1$. We define the augmented random variable for each data point in $\mathcal{I}_2$,

$$\tilde{q}_i^s := \hat{q}_i^s + U_i^s([-\sigma, \sigma]) \quad \forall i \in \mathcal{I}_2^s, s \in [K],$$

where $U_i^s$ are i.i.d. random variables, uniformly distributed on $[-\sigma, \sigma]$ for some small positive $\sigma$, and independent from all the previously introduced random variables. It serves as a smoothing random variable, for the random variables $\tilde{q}_i^s$ are i.i.d. continuous for any $P(Y|\tilde{X})$ and $\mathcal{Q}$. Otherwise, the original $\hat{q}_i^s$ might have atoms resulting in a non-zero probability to observe ties in the group $\{\hat{q}_i^s\}$ for $s = 1, \ldots, K$. This trick, also called jittering [7, 12] is often used for data visualization for tie-breaking. Using the above quantities, we build the CDF and quantile function estimators for each subgroup $s' \in [K]$ as follows,

$$\hat{F}_{q_\alpha|s'}(t) = N_s^{-1} \sum_{i=1}^{N_s} \mathbb{1}\left\{\tilde{q}_i^{s'} \leq t\right\}, \quad \text{for all } t \in \mathbb{R}, \tag{7}$$

$$\hat{Q}_{2,q_\alpha|s'}(t) = \int_0^1 \hat{F}_{q_\alpha|s'}^{-1}(v) K_h(t - v) dv, \quad t \in (0, 1). \tag{8}$$

The smoothed kernel estimator Eq.(8) was firstly proposed by Cheng and Parzen [8], where $K_h(\cdot) = K(\cdot/h)/h$ is a kernel function chosen as a probability density function that is symmetric around zero with bandwidth parameter $h > 0$.

---

[2] $\hat{q}_i^s$ depends on the quantile level $\alpha$, we suppress $\alpha$ for notational simplification.

If the quantile functions $Q_{2,q_\alpha|s'}$ is differentiable, the derivative $Q'_{s'}(t) := Q'_{2,q_\alpha|s'}(t)$ for $t \in (0,1)$ is the quantile density function [8, 45]. We hereby give an estimation bound for Eq. (8) using kernel smoothing. For this purpose, we invoke the conditions (A1) - (A3) that are needed for deducing the following proposition. They can also be found from [45] and are included in the supplementary material.

**Proposition 2.** Under conditions (A1), (A2), and (A3), we have

$$\sup_{s'} \sup_{t \in [0,1]} \left| \hat{Q}_{2,q_\alpha|s'}(t) - Q_{q_\alpha|s'}(t) \right| = O_p\left( N^{-1/2} \right), \quad s' = 1, \ldots, K.$$

The motivation for including a smoothing step is twofold: First, smoothing the quantile function eliminates the troublesomeness in defining arbitrary quantiles from the empirical one when the sample sizes of subgroups are small. Second, the proposed kernel smoothing improves second-order efficiency by alleviating the relative deficiency [15, 45].

**Remark 1.** One can utilize various kernels such as the Gaussian or Epanechnikov kernel with adaptive bandwidth for better practical performance. Other smoothing methods such as splines or local linear fitting can likewise be applied with equal effectiveness.

Consequently, for each quantile level $\alpha$, the functional synchronized quantile estimator is

$$\hat{g}_{\alpha,i} = \sum_{s'=1}^{K} \hat{p}_{s'} \hat{Q}_{2,q_\alpha|s'} \circ \hat{F}_{q_\alpha|s} \circ \tilde{q}_i^s, \quad \forall i \in \mathcal{I}_2. \tag{9}$$

The proposed estimator can be deemed as the empirical counterpart with additional randomization of the explicit fair optimal formula Eq.(6).

To conformalize the adjusted fair quantiles Eq (9), we need to compute the conformity scores $E_i$ for each $i \in \mathcal{I}_2$ that quantify the error made by the plug-in fair prediction interval $\hat{C}^g(\tilde{x}) = [\hat{g}_{\alpha_{\text{lo}}}(\tilde{x}), \hat{g}_{\alpha_{\text{hi}}}(\tilde{x})]$. The scores are evaluated on the calibration set as

$$E_i := \max\{\hat{g}_{\alpha_{\text{lo}},i} - Y_i, Y_i - \hat{g}_{\alpha_{\text{hi}},i}\}. \tag{10}$$

At the last stage, for a new data point $\tilde{X}_{n+1} = (x, s)$, and $\alpha \in \{\alpha_{\text{lo}}, \alpha_{\text{hi}}\}$, by defining

$$\tilde{q}_{1,i}^s = \hat{q}_i^s + U_i^s([-\sigma, \sigma]) \quad \forall i \in \mathcal{I}_1^s \quad \text{and} \quad \tilde{q}_\alpha(x, s) = \hat{q}_\alpha(x, s) + U([-\sigma, \sigma]).$$

We use the empirical CDF of training set [3]

$$\hat{F}_{1,q_\alpha|s}(t) := \frac{1}{|\mathcal{I}_1^s|+1} \left( \sum_{i=1}^{|\mathcal{I}_1^s|} \mathbb{1}\left\{ \tilde{q}_{1,i}^s < t \right\} + U([0,1]) \left( 1 + \sum_{i=1}^{|\mathcal{I}_1^s|} \mathbb{1}\left\{ \tilde{q}_{1,i}^s = t \right\} \right) \right) \tag{11}$$

to estimate the location $\hat{F}_{1,q_\alpha|s} \circ \tilde{q}_\alpha(x, s)$. Thus the fair quantile estimator is built as follows

$$\hat{g}_\alpha(x, s) = \sum_{s'=1}^{K} \hat{p}_{s'} \hat{Q}_{2,q_\alpha|s'} \circ \hat{F}_{1,q_\alpha|s} \circ \tilde{q}_\alpha(x, s), \forall \alpha \in \{\alpha_{\text{lo}}, \alpha_{\text{hi}}\}. \tag{12}$$

The fair prediction interval for $Y_{n+1}$ is constructed as

$$C(\tilde{X}_{n+1}) = [\hat{g}_{\alpha_{\text{lo}}}(x, s) - Q_{1-\alpha}(E, \mathcal{I}_2), \hat{g}_{\alpha_{\text{hi}}}(x, s) + Q_{1-\alpha}(E, \mathcal{I}_2)], \tag{13}$$

where $Q_{1-\alpha}(E, \mathcal{I}_2) := (1-\alpha)(1 + 1/|\mathcal{I}_2|)$-th empirical quantile of $\{E_i : i \in \mathcal{I}_2\}$ will adjust the plug-in fair prediction interval. We present the pseudo-codes of CFQP as well as the construction of $\hat{g}_\alpha$ for Eq. 9 in Algorithm 1, 2 respectively.

## 5   Theoretical results

We provide a statistical analysis of the proposed algorithm with coverage and DP-fairness guarantees in this part.

---

[3]Still, $\hat{q}_i^s$ depends on quantile level $\alpha$.

---

**Algorithm 1** Split Conformal Fair Prediction (CFQP)

---

**Input**: $\mathcal{D} = \{(X_i, S_i, Y_i)\}_{i=1}^n$; miscoverage level $\alpha \in (0, 1)$; quantile regression algorithm $\mathcal{Q}$.
1: Randomly split $[n]$ into disjoint proper training and calibration indices $\mathcal{I}_1, \mathcal{I}_2$.
2: Fit two conditional quantile functions on the training set $\{\hat{q}_{\alpha_{\text{lo}}}, \hat{q}_{\alpha_{\text{hi}}}\} \leftarrow \mathcal{Q}(\{(X_i, S_i, Y_i), i \in \mathcal{I}_1\})$.
3: Call functional Synchronization (Algorithm 2) to calculate $\{\hat{g}_{\alpha_{\text{lo}}}, \hat{g}_{\alpha_{\text{hi}}}\}$ for each $i \in \mathcal{I}_2$.
4: Compute $E_i \leftarrow \max\{\hat{g}_{\alpha_{\text{lo}}}(X_i) - Y_i, Y_i - \hat{g}_{\alpha_{\text{hi}}}(X_i)\}$ for $\forall i \in \mathcal{I}_2$.
5: Compute $Q_{1-\alpha}(E, \mathcal{I}_2) \leftarrow (1-\alpha)(1 + 1/|\mathcal{I}_2|)$-th empirical quantile of $\{E_i : i \in \mathcal{I}_2\}$.
6: For a new test point $(x, s)$, compute $\{\hat{g}_{\alpha_{\text{lo}}}(x, s), \hat{g}_{\alpha_{\text{hi}}}(x, s)\}$ through Algorithm 2
**Output**: Fair prediction interval $C(x, s) = [\hat{g}_{\alpha_{\text{lo}}}(x, s) - Q_{1-\alpha}(E, \mathcal{I}_2), \hat{g}_{\alpha_{\text{hi}}}(x, s) + Q_{1-\alpha}(E, \mathcal{I}_2)]$ for $(X_{n+1}, S_{n+1}) = (x, s)$.

---

**Algorithm 2** Functional Synchronization

---

**Input**: Calibration set $\{(X_i, S_i)\}_{i \in \mathcal{I}_2}$ or new point $(x, s)$; base quantile estimator $\mathcal{Q}$;
  slack parameter $\sigma$; training set $\{(X_i, S_i)\}_{i \in \mathcal{I}_1}$;
1: **if** Calibration set $\{(X_i, S_i)\}_{i \in \mathcal{I}_2}$ **then**
2:   **for** $\alpha \in \{\alpha_{\text{lo}}, \alpha_{\text{hi}}\}$ **do**
3:     $\{\tilde{q}_\alpha(X_i, S_i)\} \leftarrow \{q_\alpha(X_i, S_i) + U([-\sigma, \sigma])\}_{i \in \mathcal{I}_2}$       $\triangleright U([-\sigma, \sigma])$ are used for tie-breaking
4:     **for** $s' \in [K]$ **do**
5:       Compute $\hat{F}_{q_\alpha|s'}(t)$, and $\hat{F}_{2,q_\alpha|s'}^{-1}(t)$ by Eq. (7) and (8).
6:       Obtain $\hat{g}_\alpha(X_i, S_i) \leftarrow \sum_{s'=1}^K \hat{p}_{s'} \hat{F}_{2,q_\alpha|s'}^{-1} \circ \hat{F}_{q_\alpha|s'} \circ \tilde{q}_\alpha(X_i, S_i), \forall i \in \mathcal{I}_2$
7:     **end for**
8:   **end for**
9: **else if** New test point $(x, s)$ **then**
10:   **for** $\alpha \in \{\alpha_{\text{lo}}, \alpha_{\text{hi}}\}$ **do**
11:     $\{\tilde{q}_{1,\alpha}^s\} \leftarrow \{\hat{q}_\alpha^s + U([-\sigma, \sigma])\}_{i \in \mathcal{I}_1^s}$ and $\tilde{q}_\alpha(x, s) \leftarrow q_\alpha(x, s) + U([-\sigma, \sigma])$
12:     Compute $\hat{g}_\alpha(x, s) \leftarrow \sum_{s'=1}^K \hat{p}_{s'} \hat{F}_{2,q_\alpha|s'}^{-1} \circ \hat{F}_{1,q_\alpha|s} \circ \tilde{q}_\alpha(x, s)$ by Eq. (8) and (7)
13:   **end for**
14: **end if**
**Output**: fair quantile prediction $\hat{g}_\alpha$ for calibration set or new test point $(x, s)$.

---

**Theorem 1** (Prediction coverage guarantee). *If $(\tilde{X}_i, Y_i), i = 1, \ldots, n + 1$ are exchangeable, then the prediction interval $C(\tilde{X}_{n+1})$ constructed by the split CFQP algorithm satisfies*

$$P\{Y_{n+1} \in C(\tilde{X}_{n+1})\} \geq 1 - \alpha.$$

*Moreover, if the conformity scores $E_i$ are almost surely distinct, the prediction interval is nearly exactly calibrated,*

$$P\{Y_{n+1} \in C(\tilde{X}_{n+1})\} \leq 1 - \alpha + 1/(|\mathcal{I}_2| + 1).$$

**Remark 2.** Corollary 1 in the supplementary material gives an extension for the conformalization step which allows coverage errors to be spread arbitrarily over the left and right tails. Controlling the left and right tails independently yields a stronger coverage guarantee.

**Theorem 2** (Demographic parity guarantee). *For any joint distribution $P$ of $(X, S, Y)$, any $\sigma > 0$, as well as the base quantile estimator $\hat{q}_\alpha : \mathbb{R}^p \times [K] \to \mathbb{R}$ constructed on labeled data, the estimator $\hat{g}_\alpha$ defined in Eq. (12) satisfies*

$$(\hat{g}_\alpha(X, S) \mid S = s) \stackrel{d}{=} (\hat{g}_\alpha(X, S) \mid S = s') \quad \forall s, s' \in [K].$$

Quantile DP guarantees provided by Theorem 2 are derived directly from distribution-free properties on rank and order statistics in Lemma E.1., Theorem 7.2, of Chzhen and Schreuder [11]. Further information can be found in their papers and the references they provide. We extend the estimator $\hat{g}$ for quantile regression based on the estimators developed in their work with solid theoretical foundations. In contrast to the seminal work of Chzhen and Schreuder [11], we regard the densities and quantile functions as functional data, namely, as samples of stochastic processes. Typically, this approach is used by economists when dealing with the densities of income distribution across populations. As a result of introducing the kernel smoothing step, potential performance improvements are demonstrated in the supplemental material.

# 6 Experiments

To evaluate our proposed method [4], we report the performance of post-processing fairness adjustment on quantiles through four benchmark datasets: Law School (LAW), Community&Crime (CRIME), MEPS 2016 (MEPS), Government Salary (GOV). A detailed description of these datasets can be found in the supplementary material. The code for reproducing our results is avaiable at `https://github.com/Lei-Ding07/Conformal_Quantile_Fairness`.

|  | LAW | | | | CRIME | | | |
| --- | --- | --- | --- | --- | --- | --- | --- | --- |
|  | Coverage | Length | KS(lo) | KS(hi) | Coverage | Length | KS(lo) | KS(hi) |
| Ln-CQR | 90.16±0.47 | 0.46±.004 | 0.39±0.03 | 0.11±0.02 | 90.22±1.88 | 1.30±0.05 | 0.62±0.06 | 0.53±0.06 |
| **Ln-CFQP** | 90.02±0.51 | 0.46±.004 | 0.02±0.01 | 0.02±0.01 | 90.44±1.84 | 1.64±0.05 | 0.11±0.03 | 0.12±0.04 |
| RF-CQR | 90.25±0.55 | 0.39±.005 | 0.20±0.02 | 0.15±0.02 | 90.27±1.66 | 1.15±0.03 | 0.64±0.05 | 0.59±0.05 |
| **RF-CFQP** | 90.11±0.48 | 0.38±.004 | 0.02±.008 | 0.02±.009 | 90.34±1.84 | 1.54±0.04 | 0.12±0.04 | 0.12±0.03 |
| NN-CQR | 90.00±0.50 | 0.40±0.02 | 0.41±0.07 | 0.18±0.05 | 90.01±1.89 | 1.16±0.05 | 0.70±0.05 | 0.63±0.06 |
| **NN-CFQP** | 90.01±0.51 | 0.39±0.01 | 0.02±.009 | 0.03±.009 | 89.95±1.62 | 1.54±0.12 | 0.12±0.04 | 0.12±0.03 |
|  | MEPS | | | | GOV | | | |
|  | Coverage | Length | KS (lo) | KS(hi) | Coverage | Length | KS (lo) | KS(hi) |
| Ln-CQR | 89.92±0.66 | 0.66±0.01 | 0.09±0.03 | 0.33±0.05 | 90.00±0.19 | 0.79±.002 | 0.26±.014 | 0.44±0.02 |
| **Ln-CFQP** | 89.99±0.69 | 0.66±0.01 | 0.03±0.01 | 0.03±0.01 | 90.02±0.19 | 0.78±.002 | 0.05±0.01 | 0.04±0.01 |
| RF-CQR | 90.07±0.65 | 0.38±.009 | 0.19±0.02 | 0.30±0.03 | 90.03±0.17 | 0.61±.002 | 0.29±0.01 | 0.28±0.02 |
| **RF-CFQP** | 90.38±0.60 | 0.39±0.01 | 0.02±0.01 | 0.03±0.01 | 90.03±0.17 | 0.62±.002 | 0.05±0.01 | 0.04±0.01 |
| NN-CQR | 89.95±0.68 | 0.37±0.04 | 0.24±0.09 | 0.37± 0.06 | 90.01±0.19 | 0.58±0.01 | 0.28±0.03 | 0.32±0.04 |
| **NN-CFQP** | 89.97±0.61 | 0.37±0.04 | 0.03±0.01 | 0.04±0.01 | 90.01±0.18 | 0.59±0.01 | 0.05±0.01 | 0.05±0.01 |

Table 1: Results reported on test set of 200 repeated experiments with $\alpha = 0.1$. CQR refers to the conformalized quantile regression in [34]. Ln, RF, and NN denote the linear, random forest, and neural network quantile regression models. Our methods are shown in bold.

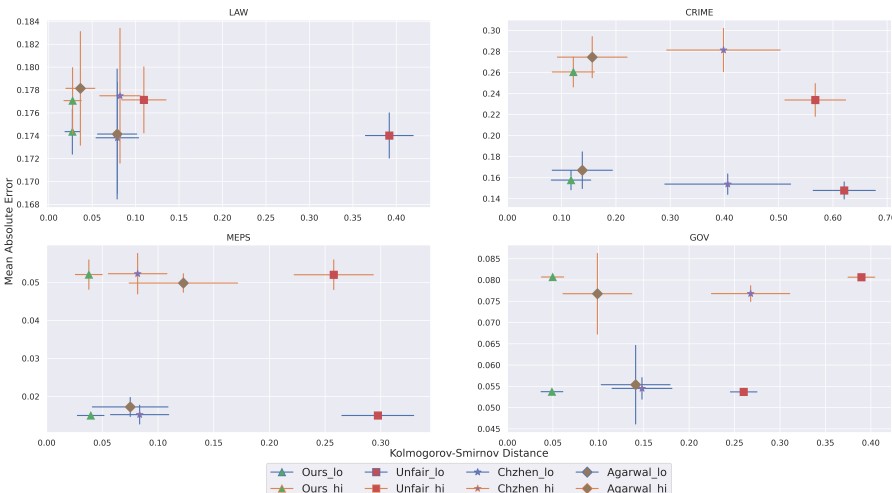

Figure 2: Results for estimating the lower ($\alpha_{\mathrm{lo}}$) and upper ($\alpha_{\mathrm{hi}}$) quantiles using some state-of-the-art DP-fairness requirement methods on all the datasets. 'Unfair', 'Chzhen', and 'Agarwal' stand for the linear quantile model without fairness adjustment, barycenter method [12] and reduction-based algorithm [1] respectively. We present the MAE and KS of lower and upper quantile estimations. A Linear quantile model is implemented in this comparison.

We measure the violation of DP-fairness of the quantiles required by Definition 1 through the empirical Kolmogorov-Smirnov (KS) distance. The value represents the disparity between groups

---

[4]We utilize the local linear fitting smoothing method in the experiments.

$$\mathcal{Z}^s = \{(X, S, Y) \in \mathcal{Z} : S = s\} \text{ for all } s \in [K],$$

$$\text{KS}(g_\alpha) = \max_{s,s' \in [K]} \sup_{t \in \mathbb{R}} \left| \frac{1}{|\mathcal{Z}^s|} \sum_{(X,S,Y) \in \mathcal{Z}^s} \mathbb{1}\{g_\alpha(X, S) \leq t\} - \frac{1}{|\mathcal{Z}^{s'}|} \sum_{(X,S,Y) \in \mathcal{Z}^{s'}} \mathbb{1}\{g_\alpha(X, S) \leq t\} \right|.$$

**Experiment results.** In Table 1, we report the average performance of the proposed CFQP over 200 randomly training-test splits as well as the baseline model CQR by the coverage rate, length of prediction interval, and the KS distance of the interval endpoint. We split the training data into proper training and calibration sets of equal sizes. Throughout the experiments, the nominal miscoverage rate is fixed to $\alpha = 0.1$. Among pre-existing quantile algorithms, we select three leading variants: **linear model**[24], **random forests** [27] and **neural networks** [35]. Overall, our CFQP likewise CQR constructs prediction bands attaining desirable coverage around 90%, as claimed in Theorem 1. Random forest based approaches tend to be slightly more conservative than the other two w.r.t the coverage rate among all four datasets.

In the KS column concerning the DP fairness of interval endpoints, our CFQP method greatly reduces the discriminatory bias (quantified by KS) by 70% up to 90% compared to that of CQR. In addition, the lengths of the prediction intervals mostly remain the same except for the Crime dataset, probably due to its inherent high discriminatory bias between sensitive groups. In addition, CFQP is more robust according to the standard errors over experiment repetitions.

Figure 2 presents the comparison of our post-processing fairness adjustment procedure on quantiles $\hat{g}_\alpha$ using the test set $\mathcal{Z}_{test} = \{(X_i, S_i, Y_i)\}_{i=1}^{n_{test}}$ with some state-of-the-art fairness algorithms. Since most of the algorithms are targeted at mean prediction, there is no direct comparison with our quantile fairness method; we accordingly modified the existing methods into quantile versions for comparison. A detailed description can be found in the supplementary material.

The points in Figure 2 represents the mean of 200 repeated experiments with $x$-axis as KS distance and $y$-axis as Mean Absolute Error(MAE), where $\text{MAE}(g_\alpha) = 1/n_{test} \sum_{\mathcal{Z}_{test}} |Y - g_\alpha(X, S)|$ measures the prediction error of quantiles, the bars are the standard error on both axes. The optimal points should locate at the bottom left corner of the graph, where smaller KS distance and smaller MAE are achieved. In each subplot, our method consistently performs better with the smallest KS distance while keeping the MAE equal or slightly higher than the others or the unfair version. Note that due to the highly right skewness of real datasets, the MAE of the upper quantile estimation is larger than that of the lower quantile for all quantile approaches. Additional ablation studies for computational time analysis and testing the kernel smoothing approach are incorporated in the supplementary material.

## 7    Conclusion and future work

Conformal fair quantile regression is a novel approach for creating fair prediction intervals that attain valid coverage and reach independence between sensitive attributes while making minimal modifications to the quantile endpoints simultaneously. It becomes superior within heteroskedastic and/or asymmetric datasets and robust to outliers.

Our method is supported by rigorous distribution-free coverage and exact DP-fairness guarantees, as proved in theoretical parts. We conducted several real data examples demonstrating the effectiveness of our method in achieving exact coverage while imposing DP-fairness in practice. The method outperforms several state-of-the-art approaches by comparison.

A limitation in our numerical experiments is that we simply utilize the local linear smoothing method in defining quantile functions of the subgroups; we believe incorporating flexible kernel smoothing approaches [40, 45] would improve the experimental performances.

As potential future works, it would be valuable to introduce a DP relaxation framework based on an unfairness measure in a similar manner as [11, 37], allowing controlling the level of unfairness in quantile estimates. We also expect to extend the scope to other potential fairness metrics which is dependent on the underlying response like equalizing quantile loss across groups by incorporating a fairness penalty term in training, or the fairness metric defined for conditional variance-at-risk. In addition, it is worthwhile exploring the quantile fairness in other areas of AI, such as NLP[14, 22], Computer Vision, Recommendation systems, etc.

## Acknowledgements

This work was supported by the Economic and Social Research Council (ESRC ES/T012382/1) and the Social Sciences and Humanities Research Council (SSHRC 2003-2019-0003) under the scheme of the Canada-UK Artificial Intelligence Initiative. The project title is BIAS: Responsible AI for Gender and Ethnic Labour Market Equality. We thank Yang Hu, Xiaojun Du and Matthew Pietrosanu for their helpful discussion and valuable input and all the constructive suggestions and comments from the reviewers.

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
