# OpenReview forum: "Conformalized Fairness via Quantile Regression"
_NeurIPS.cc/2022/Conference — NeurIPS 2022 Accept_

### Official Review · Reviewer_H28y · 2022-07-10

**Rating:** 5
**Confidence:** 2
**Soundness:** 3 good
**Presentation:** 2 fair
**Contribution:** 3 good

**Summary:**

This paper studies Demographic Parity in Quantile Regression. The authors extend from previous work in Conformalized Quantile Regression and use optimal transport and functional synchronization techniques to incorporate DP into Quantile Regression. Experiments show that the fair version of Quantile Regression significantly reduces the gap in DP while keeping a good coverage rate and length of the interval.

**Questions:**

1. It seems that the template is not in submission mode.

**Limitations:**

Limitations and negative societal impacts are adequately addressed.

**Strengths And Weaknesses:**

Pros:
1. Consider the fairness issue in quantile regression is novel. It seems that there is no other paper considering this specific problem.
2. The proposed techniques look sound and make sense to me.
3. Good experimental results. Compared to the non-fair version, the proposed method gets a small gap in terms of KS at a minimal cost of coverage rate and interval length.

Cons:
1. There is limited context characterizing the fairness issue in quantile regression. Probably the authors can describe without a careful design how unfairness can be propagated in quantile regression through a vivid example, and how such unfairness is different from other problems, e.g., ordinary regression.
2. Some key information in experiments is missing. The authors mention in Fig. 2 they compare to 'some state-of-the-art fairness algorithms' but there is no reference and discussion.

---

> ### Author Response · Authors · 2022-08-02
> **Reply to reviewer H28y**
>
> ### Response to Cons 1:
> Thanks a lot for this constructive comment. We may take an example from societal studies [1,2]. The wage gap between the two genders in Spain, in line with what happens in other countries, is quite substantive. On average, women earn around 70\% as much as men. The unemployment rate for Spanish women, at 30\%, doubles that of men. \
> \
> A large part of this difference cannot be accounted for by observable variables such as experience, sector of employment, or education, instead, the unexplained parts in salary differences between men and women have been interpreted as a degree of wage discrimination against women. Indeed, when the wage gap is computed by levels of education, the same survey reveals that women who have completed a university degree earn on average only 60\% of the salary received by men with the same educational level. Results for other countries detect the same qualitative pattern.
>
> From results in Table 5 of [1], we can definitely tell that unfairness not only exists among mean estimation but can be propagated, and even worsen in quantiles. Also, the wage gap increases with the pay scale: while the wage floor of the best-paid 50\% of men with average characteristics is estimated to be around 12\% greater than the wage floor of the best-paid 50\% of women, the wage floor for the best-paid 10\% of men is around 15\% greater than that of the best-paid 10\% of women. the results are consistent with the reported claims of more frequent and greater discrimination on behalf of women at high salary levels. Therefore, it's urgent and essential to deal with bias or unfairness in quantile estimations. We have also updated our manuscript to incorporate this example in section 1.
>
> ### Response to Cons 2:
> Thank you for the comment. We have added the reference and some implementation discussions in the supplementary material in the revised version (marked in blue).
>
> ### Response to Question 1:
> Thank you for the comment. We sincerely apologize for the partially incorrect template, which we have corrected in the resubmission.
>
>
> [1] J. García, P. J. Hernández, and A. López-Nicolás. How wide is the gap? an investigation of gender
> wage differences using quantile regression. Empirical Economics, 26(1):149–167, Mar 2001. doi:
> 10.1007/s001810000050. \
> [2] J. Gardeazabal and A. Ugidos. Gender wage discrimination at quantiles. Journal of Population Economics,
> 18(1):165–179, Mar 2005. doi: 10.1007/s00148-003-0172-z

---

> > ### Comment · Reviewer_H28y · 2022-08-08
> > **Thanks for your response**
> >
> > Thanks for addressing my concerns. I don't have any further questions and I would keep my score.

---

### Official Review · Reviewer_NTAY · 2022-07-11

**Rating:** 4
**Confidence:** 3
**Soundness:** 2 fair
**Presentation:** 2 fair
**Contribution:** 2 fair

**Summary:**

In this paper, the authors study the fairness problem in quantile regression settings. Based on the theories from the Wasserstein barycenter problem and two smoothing strategies, the authors propose the CFQP algorithm. They also theoretically analyze the results of the algorithm. Finally, they conduct experiments on four real-world datasets to prove the effectiveness of the algorithm.

**Questions:**

See details in the Strengths And Weaknesses section, especially the points in the Originality, Quality, and Clarity parts.

**Limitations:**

More limitations are demonstrated in the Strengths and Weaknesses section.


**Strengths And Weaknesses:**

**Originality**

\+ The authors propose to study the fairness problem in the quantile regression setting, which is a new problem.

\- The methods proposed by the authors seem a combination of the Wasserstein barycenter problem studied in [10] and two smoothing strategies in [6][7], which makes the contribution marginal.

**Quality**

\- The advantages of the two smoothing strategies are not clear. On the one hand, more discussions about the two smoothing strategies are needed. For example, why the jittering strategy works when the sample sizes of subgroups are small? And what are the second-order efficiency and relative deficiency brought by the kernel smoothing strategy? On the other hand, the authors are encouraged to conduct the ablation study by testing the experimental results if removing either or both of the smoothing strategies.

**Clarity**

\- The paper is not well-written. Several unclear parts need further clarification. There are also several typos to fix.
  1. The three conditions in Section 5 are confusing. What do these conditions mean and are they practical? More discussions are needed.
  2. As mentioned above, the advantages of the two smoothing strategies should be clarified.
  3. In Section 2, it is not clear what the subscript $n+1$ means since there are $n$ training samples. I understand this may refer to the unknown test sample, but please clarify it.
  4. Figure 1 caption: "affacted" --> "affected", "quartile" --> "quantile".

**Significance**

\+ The method is generally sound with both theoretical and experimental results.

---

> ### Author Response · Authors · 2022-08-02
> **Reply to reviewer NTAY.**
>
> ### Response to Originality
> Thank you for the comment. The writing in the original paper may have caused you to misunderstand the main contributions of the paper. Therefore, the entire manuscript, including the introduction and presentation of the methods, has been carefully rewritten and significantly improved. We would appreciate it if you could take a second look at the revised paper and determine what our paper’s focus and major contributions are. The main contributions are re-written and summarized in the Contributions and Outlines of the
> Introduction Section.
>
> In fact, the combination of the Wasserstein barycenter problem and two smoothing strategies
> is indeed one of our many contributions. The main purpose of this paper is to propose a new
> quantile-based method with a valid inference that enhances both accuracy and fairness while
> maintaining a balance between the two. It is a novel framework that allows exact control of
> prediction miscoverages while ensuring quantile fairness simultaneously. According to our
> knowledge, no valid statistical inference has been presented in a similar context before.
> For your convenience, we also put them here:
> 1. We successfully transform the problem of searching quantiles under DP fairness to
> the construction of multi-marginal Wasserstein-2 barycenters via the optimal transport
> theory. We incorporate a novel kernel smoothing step into the preceding method, which
> is particularly advantageous for subgroups whose sample sizes are too small to obtain
> reliable quantile function estimations.
> 2. We propose a conformalized fair quantile regression prediction interval (CFQP) inspired
> by the works of Romano et al. (2019). It is mathematically proved to achieve a finite
> sample, distribution-free validity, demographic parity on different quantiles, and exact control of miscoverage rates, regardless of the quantile algorithm used. The
> theoretical validity of the prediction interval constructed by CFQP and the exact DP of the fair
> quantile estimators are given in Section 5 and the supplement.
>
> ### Response to Quality
> Thanks a lot for the comment. We'd argue that the two smoothing approaches are crucial in the success of constructing the DP-fairness quantile estimators and deriving the conformal fair prediction interval.
>
> 1. Note that the quantile fairness guarantee given in the theoretical results does not necessitate additional assumptions about either the base quantile estimator $q_\alpha$ or the joint distribution ${P}$ of $(X, S, Y)$. This is because of the jittering step in the definition of $\hat{g}_\alpha$ in Eq.(11) where we artificially introduce continuity into the formulation. Due to continuity, we can utilize the conclusions from the theory of rank statistics of exchangeable random variables [ 5,6,14]  to obtain Eq.(11). The exact DP guarantee in Theorem 2 is likewise distribution-free and can be applied on top of any base predictor since the fundamental results on rank statistics as long as the underlying random variable is continuous.
>
> 2. The kernel smoothing quantile estimator serves as an effective tool for varying the quantile function estimates. Note that the proposed CFQP approach which gives valid prediction intervals with fair endpoints (quantiles) requires 'almost surely distinct' conformity scores $E_i$ (Theorem 1) in order to attain the nearly exactly calibrated prediction interval. Theorems have shown that the kernel smoothing quantile estimators are better options than their empirical counterparts  [ 4, 16 ]. On the one hand, smoothing reduces the random variation in the data, resulting in a more efficient estimator, which is the case under subgroups of small sample sizes.
> In particular, the second-order efficiency and relative deficiency [2,9] in statistics are measures to compare different asymptotic estimators: $\hat{Q}$ and $\hat{F}^{-1}$ in this work. The second-order efficiency and relative deficiency are the inverses of the sample sizes required to obtain the same mean square error (based on the population quantile function).  The kernel smoothing quantile estimator requires less sample in practice to attain accurate estimation. On the other hand, smoothing also gives a smooth curve for the quantile function $\hat{Q}$ (Eq.(8)) that better displays the intrinsic features of the group population distribution.
>
> 3. We also conducted a brief ablation study by testing the experimental results of removing either or both of the smoothing strategies in our CFQP approach. Results are presented in the supplementary material. We found incorporating the jittering and kernel smoothing methods works better when the subgroups are unbalanced and there exist subgroups of small sample sizes, especially for the CRIME dataset.
>
> **Please see the following reply for the response to Clarity and references.**

---

> > ### Author Response · Authors · 2022-08-02
> > **Response to Clarity and references.**
> >
> > ### Response to Clarity.
> >
> > 1. First of all, we sincerely apologize for the latex format, and the presentation of the previously submitted manuscript, which may affect your impression and reading of this paper. We have rewritten sections 1 and 2, together with several other unclear parts (marked in blue) in the resubmission. We would appreciate it if you could take a second look. We have also carefully reviewed the work for typos and missing definitions.
> >
> > 2. In addition, we provide a detailed explanation of the aforementioned three conditions of Section 5 in the supplementary material. We also state them here for your convenience:
> >
> > - Conditions (A1)-(A3) guarantee the existence of a strong approximation of the empirical quantile process by a sequence of weighted Brownian bridges as established in Csorgo and Revesz (1978). Condition (A3) posited on kernel functions assures that the integral transform $\hat{F}^{-1} \mapsto \hat{Q}$ possesses good approximation properties for smooth functions, and it is shown that (A3) holds for any difference kernel $\mathrm{d}{t}K_{h}(u, t)=h^{-1} k\left((u-t)/h\right)\mathrm{d}t$ with a vanishing bandwidth $h$. For example, the gaussian density $K(u)=\exp(-{u^{2}}/{2 h^{2}})$  or the triangular density function $K(u)=(1-|u|/h)I(|u|/h\le1)$ with a vanishing bandwidth $h_n$.
> >
> > References:
> >
> > [2] P. J. Bickel, D. M. Chibisov, and W. R. Van Zwet. On Efficiency of First and Second Order, page 185–191.
> > 2012. doi: 10.1007/978-1-4614-1314-1_12.
> >
> > [4] C. Cheng and E. Parzen. Unified estimators of smooth quantile and quantile density functions. Journal of
> > statistical planning and inference, 59(2):291–307, 1997.
> >
> > [5] E. Chzhen and N. Schreuder. A minimax framework for quantifying risk-fairness trade-off in regression.
> > arXiv:2007.14265 [math, stat], Jan 2022. arXiv: 2007.14265.
> >
> > [6] E. Chzhen, C. Denis, M. Hebiri, L. Oneto, and M. Pontil. Fair regression with wasserstein barycenters.
> > Advances in Neural Information Processing Systems, 33:7321–7331, 2020.
> >
> > [9] J. K. Ghosh and K. Subramanyam. Second order efficiency of maximum likelihood estimators. Sankhy ̄a:
> > The Indian Journal of Statistics, Series A (1961-2002), 36(4):325–358, 1974. ISSN 0581572X.
> >
> > [14] V. Vovk, A. Gammerman, and G. Shafer. Algorithmic learning in a random world. Springer, New York,
> > 2005. ISBN 978-0-387-00152-4.
> >
> > [16] S.-S. Yang. A smooth nonparametric estimator of a quantile function. Journal of the American Statistical
> > Association, 80(392):1004–1011, 1985. ISSN 0162-1459. doi: 10.2307/2288567.

---

> > > ### Comment · Reviewer_NTAY · 2022-08-08
> > > **Response to authors**
> > >
> > > I appreciate your detailed reply. However, my concerns about the novelty of the method remain. In my opinion, the method is based on the result proposed in [1] and the authors provide several estimators to approximate different parts in Equation (6). However, it is not clear what is the contribution compared with the estimating methods proposed in [1]. For example, Equation (6) in [1] also adopted a jittering technique similar to this paper.
> > >
> > > [1] Chzhen, Evgenii, et al. "Fair regression with wasserstein barycenters." Advances in Neural Information Processing Systems 33 (2020): 7321-7331.

---

> > > > ### Author Response · Authors · 2022-08-08
> > > > **Response to Reviewer NTAY**
> > > >
> > > > Dear reviewer,  thank you for the comment. We did not claim Equation (6) as our contribution, it is just a technique that we modified and applied to achieve fairness. \
> > > > Among our core contributions:
> > > >
> > > > 1. The paper aims to focus the **fairness on quantile** other than just the mean prediction, which has a more profound **social impact** on fairness.
> > > >
> > > > 2. It provides a **conformalized confidence interval** that enables **statistical inference** of the fair algorithms as well as rigorous **theoretical guarantee**.
> > > >
> > > > To our best knowledge, none of these points have ever been investigated in the fairness literature. Please kindly reconsider our paper in light of these two perspectives.

---

> > > > > ### Comment · Reviewer_NTAY · 2022-08-09
> > > > > **Response to authors**
> > > > >
> > > > > Thanks for the authors' response. My concerns about the novelty of the method remain. I understand the authors study a different problem compared with [1]. However, the method proposed in this paper seems a direct application of that in [1]. As a result, could the authors highlight the technical contribution compared with [1]?
> > > > >
> > > > > [1] Chzhen, Evgenii, et al. "Fair regression with wasserstein barycenters." Advances in Neural Information Processing Systems 33 (2020): 7321-7331.

---

> > > > > > ### Author Response · Authors · 2022-08-09
> > > > > > **Response to Reviewer NTAY**
> > > > > >
> > > > > > Thank you very much for your comment. We appreciate your response though we do not agree with it.  The problem we deal with is novel and challenging by itself; which has great social impact and motivated us to attack it. Our framework is not a direct application of that in [1]. The method in [1], while being one ingredient of our framework, is not capable of achieving our goal by itself.
> > > > > > Besides that in [1], other important technical contributions are quantile estimation with novel kernel smoothing, conformal inference, and combining different parts together with a valid theoretical guarantee.
> > > > > >
> > > > > > Only by combining these technical tools, we are able to achieve the goal of providing fairness quantile estimate and fairness statistical inference through prediction interval.
> > > > > >
> > > > > > Apart from the technical aspect, please also kindly consider the social impact of our contribution which is also an essential point in the fairness paper.
> > > > > >
> > > > > > Please kindly reconsider our paper, thank you.

---

> > > > > > > ### Comment · Reviewer_NTAY · 2022-08-09
> > > > > > > **Response to authors**
> > > > > > >
> > > > > > > Thanks for the authors' response. I am satisfied with the problem setting while I still believe the novelty of the method is marginal. The motivating theoretical result (Proposition 1) and the kernel smoothing are proposed in previous works. As a result, I will keep my score unchanged.

---

> > > > > > > > ### Author Response · Authors · 2022-08-09
> > > > > > > > **Response to Reviewer NTAY**
> > > > > > > >
> > > > > > > > Thank you for the comment. We are glad to learn that you agree with the novelty of the problem
> > > > > > > > setting, and are satisfied with it.
> > > > > > > >
> > > > > > > > We would like to summarize our rebuttal with you. First, as you are not satisfied with the clarity
> > > > > > > > and the presentation of the previous submission, we put great effort and tried our best to improve it.
> > > > > > > > The manuscript has been improved a lot, which has been confirmed by you and reviewer 3BdP who
> > > > > > > > affirmed that "This work addresses an important problem via a principled approach. Thus, moving
> > > > > > > > the fair learning field forward beyond notions based on averages to quantiles".
> > > > > > > >
> > > > > > > > Second, after several rounds of fruitful interaction with you, you have agreed with the novelty of our
> > > > > > > > problem setting. It has a great social impact and is indeed a societal issue urgently to be tackled. We
> > > > > > > > appreciate that you are now satisfied with it.
> > > > > > > >
> > > > > > > > Third, we do not agree with your comment that the novelty of our method is marginal. The innovations
> > > > > > > > and novel methods proposed by AI researchers are mostly, if not all, based on the existing technical
> > > > > > > > tools; for instance, the method in Chzhen, et al. [1] is based on the barycenter theorem, a renowned
> > > > > > > > theoretical result from optimal transport. To our best knowledge, utilizing kernel smoothing to
> > > > > > > > approximate quantile functions has not been proposed previously in the social fairness field.
> > > > > > > > Our proposed approach innovatively combines several techniques, including the Wasserstein barycenter theorem, kernel smoothing, and conformal inference, as a comprehensive framework to solve a
> > > > > > > > novel problem.
> > > > > > > >
> > > > > > > > Additionally, we do not agree that our theoretical results (Section 5) have been proposed in previous
> > > > > > > > works. The derivation is not straightforward and requires challenging and rigorous proofs. They are
> > > > > > > > deliberately derived to provide theoretical guarantees in this new and novel setting.
> > > > > > > >
> > > > > > > > In summary, we believe we have successfully addressed all your initial and current concerns and
> > > > > > > > comments. The problem we are dealing with has a large social impact; it is novel and challenging by
> > > > > > > > itself; The approach we proposed is also comprehensive and innovative.

---

### Official Review · Reviewer_kfJM · 2022-07-17

**Rating:** 6
**Confidence:** 3
**Soundness:** 3 good
**Presentation:** 3 good
**Contribution:** 3 good

**Summary:**

This work investigates the incorporation of group-fairness into the established learning problem of conformalized quantile regression. In particular, the authors extend previously developed quantile regression techniques to handle demographic parity constraints between groups. To do this,  the authors develop a postprocessing technique for modifying a given quantile regressor such that it becomes fair with respect to demographic parity. Within this framework the authors provide theoretical gurentess with respect to the efficacy of their method. In particular Theorems 1, 2 demonstrate that the proposed technique produces a model which has both reasonable predictive power as well as fairness. Further, the authors compare their technique to two other fair regression methods and show that their method iid frequently superior in terms of fairness and error.



**Questions:**

## Questions and Comments


- **Slack in fairness**: In practice achieving perfect fairness (i.e. equal DP across all groups) is infeasible or at the very least undesirable. As such it is useful to have a so called "tunable slackness variable" $\varepsilon$, such that fairness can be framed as $|M_i - M_j| \leq \varepsilon$ between any two groups $i, j$, where $M$ is a statistical metric applied to both groups. The current model requires $\varepsilon=0$. Can we say anything theoretically about when $\varepsilon > 0$. Similarly, is a model designer able to control the desired unfairness (i.e. $\varepsilon$) in the practice? If so, are there any experimental results on error-fairness tradeoffs as we have vary $\varepsilon$?

- **Equation 2**: Equation 2 seems to be misspecified compared to [33]. Also the suppression of the dependence that $q$ has on $x$ and $\alpha$ in this definition may lead to confusion. It may be more clear to use $q_{\alpha}(x)$ instead of simply $q$, especially since $q$ is defined to be a function two lines earlier.

- **Citations for legal justifications**: Legal justification is given for both demographic parity as well as choice of sensitive feature S. In both instances the authors cite computer science papers (not legal works). Although the claims are mentioned in these citations, it would be better to directly cite the actual source, thus avoiding any whisper-down-the-lane issues that may arise.

- **Distribution notation**: The variable $\mathbb{P}$ is used to denote distributions. Perhaps $P_{X}$, $D$, or $\mathcal{D}$ would be more standard. Note that  [33, 41] do not use $\mathbb{P}$ to denote distributions.

- **Interpretation of DP**: Prior to the definition of demographic parity, the authors mention that demographic parity can be thought of as a stronger version of the so called "four-fifths rule". While this is a reasonable interpretation for DP in the context of classification, this example does not necessarily translate to regression. For example in a regression problem what does it mean for a candidate to "receive a job"?. In [1] (the paper from which this interpretation is pulled) the authors are quite careful to point out that they are referring to DP in the context of classification during this example. This distinction should either be made more precise, or an alternative interpretation should be given.


- **Comparison to other fair models**: In the experiments related to Figure 2, the authors mention that the models they compare two are not quantile regressors so they adapt these models to the quantile regression paradigm. Can the authors give a more percicse description of this adaptation procedure? Is this as simple as altering the labels in the training set, then running the original learning scheme, or is there something else being done? Additionally, the comparisons in this figure are for linear models only do we see similar results for non-linear models?


- **Minor comments/issues**:

    - NeurIPS latex format appears to be partially incorrect (no space for authorship and line numbers are missing).

    - The first sentence of the abstract “Fairness and impartiality have been among the predominant pursuits in human society” comes across as a slight exaggeration. One could make the argument that fairness is more often than not, treated as an afterthought rather than primary goal, i.e. ensuring systems are just fair enough to be acceptable for the time. As such I think some readers are likely to take issue with this claim as-is.

    - Page 1, second to last line: "categories ... would" should have a common instead of \dots

    - Page 2, last paragraph: The first sentence "While current approaches ... " is incomplete.

    - Page 3, second paragraph: "$\mathbb{E}$ and $\mathbb{P}$ represent for the generic ..." should be "$\mathbb{E}$ and $\mathbb{P}$ represent the generic ...". Also this sentence starts with math notation.
    - Page 3, Section 3, first paragraph: "... since then it becomes more and more pervasive...", should be "... since then it has become more pervasive ..."
    - Page 3, Section 3, first paragraph: "As a brief introduction to the quantile regression", this sentence is incomplete and also should not have "the" before "quantile regression".

    - Definition 1: It is more standard to give these definitions as the difference between a sententive group and the population, rather than between two group, e.g. $|M_g - M|$ for all groups $g$, where $M_g$ is the statistical metric applied to $g$ and $M$ is the statical metric applied to the whole population.

    - Page 4, last paragraph: "Proposition 1 pronounces", pronounces is out of place, perhaps "implies" would be better.

    - Page 7, Remark 2: "...tails as Corollary 1 in Appendix", missing "the" before "Appendix".

    - Page 6, last paragraph: "...presented in Lemma 3in the ...", missing a space between "3" and "in".

   - Page 8, Government Salary: "The yearly salary for over 200 hundred examples...", I believe this should be "The yearly salary for over 200,000 examples...", based on the size of gov_census.rda in the supplement.

    - Figure 2: It may be better to set the minimum of the $x$-axis to $0$.

     - Inconsistent use of "supplement" and "appendix" throughout the paper.



**Limitations:**

The authors have adequately addressed the limitations of their work.

**Strengths And Weaknesses:**

## Strengths
- The paper is well written (barring a few minor issues mentioned later).
- The authors make non-trivial contributions to a well established problem, namely introducing a notion of group fairness to the quantile regression problem.
- A mix of theoretical experimental results are provided.
- The proposed method is shown to outperform two state-of-the-art methods in experiments.
- The authors make a point to address the practicality of their method (e.g. Remark 1).

## Weaknesses
- The experiments performed are somewhat limited.
- Running time and computability of certain quantities appears to be missing. What is the worst case running time of Algorithm 1. How feasible is it to compute equation 8 (does this have a closed form solution or must we rely on numerical approximations)?
- The proposed model only works with one fairness metric (namely DP). It would be nice to see similar results for a fairness metric which is dependent on the underlying $Y$.
- Allowed unfairness does not appear to be a tunable parameter.
- See Questions section for more details comments.

---

> ### Author Response · Authors · 2022-08-02
> **Reply to reviewer kfJM.**
>
> We would like to thank the reviewer for your constructive feedback and your attention to detail when reviewing our work. For convenience, changes made in our revised paper are noted in blue.
>
> - Response to **Slack in fairness**:
> \
> We are with the reviewer that perfect fairness may not be necessary for applications, a "tunable slackness" variable would make our prediction more flexible by balancing accuracy and the fairness constrain. As a possible future work to the current quantile fairness framework, we could introduce a novel DP relaxation approach based on an unfairness measure. Inspired by the related work of  [5, 15], we say a quantile predictor $q_\alpha$ satisfies the $\varepsilon$-relative fair (RF) constraint for some $\varepsilon \in[0,1]$ if its unfairness is at most an $\varepsilon$ fraction of the unfairness of the original regression function $q_\alpha$, that is, $\mathcal{U}(g_{\alpha,\varepsilon}) \leq \varepsilon\mathcal{U}\left(q_\alpha\right)$.The criterion for fairness is proposed relative to the unfairness degree of $q_\alpha$, which enables a more informed choice of $\varepsilon$.\
> \
> In the current work, we have shown that the optimal fair quantile predictor $g_\alpha$ takes the form of Eq.(6).
> Then based on $g_\alpha$, the family of oracle $\varepsilon$-RF quantile estimator $g_{\alpha,\varepsilon}$ for $\varepsilon\in[0,1]$ could be derived as point-wise convex combinations of the regression function $q_\alpha$ and the optimal fair predictor $g_\alpha$. we intend to explore this respect theoretically and experimentally for future work.
>
> - Response to **Equation 2**:  \
>  Thank you for your attention to this important detail.  we have corrected it.
>
> - Response to **Citations for legal justifications**:  \
>  Thank you for the suggestions. In the revised manuscript, we have added the legal justifications from the actual source.
> - Response to **Distribution notation**:  \
> Thank you for catching this.  We have adjusted our notations.
> - Response to **Interpretation of DP**:  \
> Thank you so much for catching this, we have omitted this statement to prevent further ambiguity.
>
> - Response to **Comparison to other fair models**:  \
> Thanks a lot for the detailed comment, We have reflected this comment by adding the description in the supplement. We modified the existing methods accordingly into quantile versions for comparison.
> \
> Firstly, our approach is built upon Chzhen et al. [6]'s method, we incorporated the kernel smoothing procedure in quantile estimation and applied the local linear smoothing method provided in *NumPy* which shows its functionality when there are subgroups of small sample sizes, especially for the CRIME dataset. It is also feasible to compute Eq.(8) via some global kernels, such as the gaussian density function $K(u)=\exp(-{u^{2}}/{2 h^{2}})$  [4,16,17] or the triangular density function $K(u)=(1-|u|/h)I(|u|/h\le1)$ with a vanishing bandwidth $h_n$. For the bandwidth selection, there is a publicly available R package *lokern* with the global bandwidth choice.
> \
>  When applying the reduction-based approach in Agarwal et al. [1], we rescale and discretize the responses, and modify their algorithm by replacing the loss function $l$ with the check loss $\rho_\alpha$ in order to obtain the quantile estimation. In addition, we used logistic regression and SVM classifiers in parameter tuning.
>  \
>  In addition, we add the comparison results for non-linear models in the supplementary material. As for computational time, our approach excels a lot. For the MEPS dataset using the linear model, our method's average running time is around 20s, while it takes around 68s and 113s for the methods of  Chzhen et al. [6] and Agarwal et al. [1], respectively.
>
> - Response to **Minor comments/issues**:  \
>  We thank the reviewer for the attention to detail and thorough suggestions, all the typos and grammar problems have been corrected.
>
> References:
>
> [1] A. Agarwal, M. Dudík, and Z. S. Wu. Fair regression: Quantitative definitions and reduction-based
> algorithms. In International Conference on Machine Learning, PMLR, 2019.
>
> [4] C. Cheng and E. Parzen. Unified estimators of smooth quantile and quantile density functions. Journal of
> statistical planning and inference.
>
> [5] E. Chzhen and N. Schreuder. A minimax framework for quantifying risk-fairness trade-off in regression.
> arXiv:2007.14265 [math, stat], Jan 2022.
>
> [6] E. Chzhen, C. Denis, M. Hebiri, L. Oneto, and M. Pontil. Fair regression with wasserstein barycenters.
> Advances in Neural Information Processing Systems, 2020.
>
> [15] R. Williamson and A. Menon. Fairness risk measures. In International Conference on Machine Learning,
> PMLR, 2019.
>
> [16] S.-S. Yang. A smooth nonparametric estimator of a quantile function. Journal of the American Statistical
> Association, 80(392):1004–1011, 1985. ISSN 0162-1459.
>
> [17] Z. Zhang and H.-G. Müller. Functional density synchronization. Computational Statistics & Data Analysis,
> 55(7):2234–2249, 2011.

---

> > ### Comment · Reviewer_kfJM · 2022-08-08
> > **Response to authors**
> >
> > Thank you the response and clarification. The revised version addresses almost all my comments.
> >
> >
> > I am still curious about the actual running time of Algorithm 1, and in particular equation 8. When you say " It is also feasible to compute Eq.(8) via some global kernels, such as ...", are these the actual techniques used when computing this equation for Algorithm 1? If so, which of those kernels are you using? Moreover, it may be helpful to have these methods more verbosely explained in the supplement. Additionally, I may have missed this, but I did not see anything pertaining to the running time of the authors' proposed method. It would be good to see a theoretical worst case running time for the proposed algorithm, as well as the actual empirical running times of the authors' method and the methods used for experimental comparisons.
> >
> >
> > The lack of a tunable "fairness knob" $\varepsilon$, holds back the practicality of the paper. In practice the ability to control fairness (directly or indirectly) is important when  training models for which have both fairness and high predictive efficacy. However, I understand that the authors are investigating a somewhat new paradigm of fairness and thus asking for tunable fairness in addition to the authors' other results, may be too much. Allowing for slack in the fairness constraints sounds like a good direction for future work.

---

> > > ### Author Response · Authors · 2022-08-09
> > > **Response to reviewer kfJM**
> > >
> > > Thank you so much for the feedback.
> > >
> > > - In the current experiment, the kernel we used is the local linear one in defining the quantile functions of subgroups rather than the global kernels as we mentioned in the previous reply. In short,  when calculating the $\tau$-th quantile $x_\tau$ of $q_\alpha$ using local linear smoothing, we choose a constant distance size $h$ (kernel radius), and compute a weighted average for all data points that are closer to $x_\tau$  (the closer to $x_\tau$ points get higher weights). The time complexity for computing the local kernel smoother (Eq.(8)) is $\mathcal{O}(1)$, while if we applied the global kernels in Eq.(8), $\mathcal{O}(n)$ time would cost.\
> > > However, the main steps to determine the time complexity of Algorithms 1 and 2 reside in two parts:
> > >
> > > 1. In the for-loop we perform a post-processing which takes $\sum_{s^\prime \in [K]} \mathcal{O}\left(N_{s^\prime} \log N_{s^\prime}\right)$ time, as we need to sort the grouped samples.
> > >
> > > 2. The evaluation of $\hat{g}_\alpha$ on a new point $(x, s)$ is performed in log(sample) time as it involves locating $\hat{g}_\alpha$ in a sorted array.\
> > > We also added a more detailed explanation and computational time analysis and empirical running time to the updated version of supplementary material.
> > >
> > > - We really appreciate your advice on the tunable "fairness knob",  which is indeed compatible with our framework. We have already put this key point into our future work.
> > >
> > > Finally, thanks a lot for your detailed review and constructive feedback that greatly improve our paper.

---

### Official Review · Reviewer_3BdP · 2022-07-24

**Rating:** 7
**Confidence:** 3
**Soundness:** 3 good
**Presentation:** 3 good
**Contribution:** 3 good

**Summary:**

The work proposes a method to guarantee fair prediction intervals in regression tasks. It takes the approach of quantile regressions to obtain the prediction intervals and uses a previous result on the form of the fair quantile regressor to develop an approach for fair prediction intervals. Additionally, it uses conformal inference approach to provide intervals that have valid coverage in finite samples. The experiments on 4 real datasets show that the method maintains competitive coverage, length of prediction intervals, and prediction error compared to baselines while reducing unfairness.

**Questions:**

Questions:

Regarding presentation:

1. The relation between DP fairness defined for the interval prediction function (Def. 1) and DP fairness defined for quantile functions (between \nu’s) is not clear. How does one ensure another?

The importance or challenge in building a method that gives both fair quantile regression and conformal prediction intervals is not presented convincingly.

Regarding related work:

2. How does the proposed method for fair quantile regression relate to the method in Agarwal et al. [1]? Their method also covers the case of quantile regression with check loss and DP fairness. Does the addition of the split-conformal step applicable to their method as well to get miscoverage guarantees?

3. How does the work differ from Romano et al. [42]? Is the difference due to the fairness objective?

Regarding experiments:

4. The abstract claims that the experiments uncover the fairness vs coverage frontiers. Table 1 reports the coverage and fairness metrics but does not show the frontier. Does the claim refer to Figure 2 which reports the frontier of fairness vs error.

---
Suggestions (not expecting a response):

I would suggest tightening up the motivations for fairness (between group-specific quantile regression functions) and uncertainty estimates in the Introduction (first page). It does not clearly answer why these are required. For example what is the issue with heteroscedastic data or in the case of uncertainty estimates, do the high-stakes areas require prediction intervals or confidence intervals.

Please describe why it is challenging to combine fair regression functions with conformal inference such that the significance of the proposed method and analysis can be conveyed.

The example of salary prediction in Introduction is unclear. What is the fairness objective?

The \alpha_lo and \alpha_hi needs to be explained at page 4, first equation. Do these refer to upper and lower quantiles, \alpha/2 and 1-\alpha/2 for a miscoverage rate of \alpha?

The description of the method can be improved. Consider giving a sketch of the method before hand before describing each of the steps.

Consider discussing other potential fairness metrics in case of real-valued predictions like equalizing calibration metrics across groups.

Discuss related work Williamson and Menon 2019, Fairness risk measures. The fairness metric defined for conditional variance-at-risk is another possibility to go beyond mean predictions.
https://proceedings.mlr.press/v97/williamson19a.html

Please define DP fairness briefly to give context to Introduction and Related work.

The left hand side of Proposition 1 that is the squared loss term is not clear.

A tip on writing style is to limit the use of superlatives like tremendous as they are not precise and redundant sometimes.

**Limitations:**

Limitations are adequately discussed. Consider discussing whether the method allows for relaxing the fairness constraint that is the KS distance can be less than some non-zero value.

**Strengths And Weaknesses:**

Strengths
- addresses a new problem.
- the work seems to be important given the importance of providing valid prediction intervals in safety-critical settings while ensuring some form of fairness.
- skilfully combines advances from fair quantile regression and conformal inference.

Weaknesses
- both the problem definition and motivation is not clear from the Introduction.
- there seems to be a discrepancy between the fairness goal and the method. The aim is to define fairness of prediction intervals but the fairness metric is defined for the quantile regression functions which are one of the ways of constructing the intervals but not the only way.
- presentation is not easily understandable to readers not familiar with quantile regression-based prediction intervals and conformal inference.

The main concern is the presentation of the method. The theoretical analysis seems sound but I am not familiar with the techniques and literature so I cannot comment on its validity and significance. In my opinion, improving the presentation would take substantial changes to the writing than is possible to verify from the author response.

---

## After the response

The response adequately addresses my concerns. The presentation of the problem statement and overview of the method has been greatly improved. I appreciate the honest response to comparison with related work [1] and meticulous comparison in running time.

The work addresses an important problem via a principled approach. Thus, moving the fair learning field forward beyond notions based on averages to quantiles. Hence I have increased my score to 7, Accept.

---

> ### Author Response · Authors · 2022-08-02
> **Reply to reviewer 3BdP.**
>
> We want to thank the reviewer for all the constructive suggestions. The entire manuscript, including the introduction and presentation of the methods, has been carefully rewritten and significantly improved. It would be appreciated if you would take a second look at the revised paper (revisions are marked in blue) and see how your concerns have been addressed.
>
> ### 1. Response to question1 (regarding presentation):
> - Thank you very much for pointing out the problems with the presentation. We explain the definitions as follows. In this paper, we considered the fairness definition of DP (Def. 1) for quantile estimations since we deem quantile unfairness an important issue that requires urgent attention.  In fact, several societal studies suggest that unfairness may not be uniformly distributed across different quantiles [3,7,8,11]. We think that the DP fairness defined for quantile functions (between $\nu$’s) refers to the Wasserstein distance defined in Def. 2, and Proposition 1. They are the methodologies and solutions we employed to achieve quantile DP fairness, by assuming an optimal transportation problem and using the Wasserstein barycenter formula (Eq.(6)). In particular, Eq.(6) is a renowned result that could be used to guarantee independence between the adjusted quantile estimate $g_\alpha(X,S)$ and sensitive variable $S$, thus attain DP fairness. Intuitively, $g_\alpha(X,S)$ serves as a sample distribution mean of $[K]$ sensitive subgroups $q_\alpha^s, s=1,\dots, K$. An l2 form (mean squared deviation) is used by Eq.(5) of Proposition 1. We may utilize the l1 form (mean absolute deviation) on the l.h.s of Eq.(5) for treating $g_\alpha(X,S)$ as a distribution median in the future.
>
> ### 2. Response to questions 2 and 3 (related work):
> - Thank you for the comments. Both our barycenter-based fairness adjustment and the reduction-based method proposed in Agarwal et al. [1] with check loss applied, can provide fair quantile estimates. But these are two distinct methods using different techniques. This paper uses a Wasserstein barycenter-based approach that adequately applies the continuous property of the response, whereas Agarwal et al. [1] proposed to divide the continuous response into small groups and transform it into a multiclass classification problem, which is sensitive to how we divide it.
> - In section 6, we compared our method with that of Agarwal et al. [1] in Figure 2 and found that our method performed better in terms of fairness adjustment.  In addition, our approach excels a lot in computational time. For the MEPS dataset using the linear model, our method's average running time is around 20s, while it takes around 68s and 113s for the methods of Chzhen et al. [12]  and Agarwal et al. [1], respectively.
> - It is indeed true that the additional split-conformal step is applicable to their method to obtain miscoverage guarantees, which in turn illustrates the generalization capability of our CFQP approach.
> - In fact, what differentiates us from Romano et al. [42] is our fairness objective. In our study, we focus on DP-fairness quantiles and propose to utilize them for building prediction intervals, whereas they concentrated on equalized interval coverage.
>
> ### 3. Response to question 4 (Experiments):
>
> - Thank you very much for pointing that out. ''Frontier'' is indeed an ambiguous statement. Therefore, we revise the entire sentence and it now states that **Our results demonstrate the model's ability to uncover the mechanism underlying the fairness-accuracy trade-off in a wide range of societal and medical applications.** Accuracy is measured by the length of the prediction interval: the shorter the interval, the more accurate the prediction; vice versa. As demonstrated in Table 1 and Figure 2, our model achieves a similar level of accuracy while being substantially fairer than the other methods.
>
> **Please see the following reply for the response to the suggestions and references.**

---

> > ### Author Response · Authors · 2022-08-02
> > **Response to the suggestions.**
> >
> > ### 4. Response to suggestions:
> >
> > - We want to thank the reviewer again for the constructive and thorough suggestions. It would be deeply appreciated if you
> > could take a second look at the revised paper and see how your suggestions have been accommodated in the revised manuscript.
> > - In particular, we agree with the reviewer that it’s fascinating to consider other potential fairness metrics in the context of real-valued predictions like equalizing quantile loss across groups by incorporating a fairness penalty term in training, or fairness metrics such as conditional variance-at-risk in Williamson and Menon [15]. Therefore, we devote discussions to each in Sections 1 and 7.
> >
> > - We would like to stress again the significance and challenge of our paper. Firstly, quantile makes much better sense than mean in thinking about and studying wage gap and penalty. For example, the
> > gender wage gap and the motherhood penalty tend to vary considerably across the wage distribution.
> > However, to the best of our knowledge, little literature emphasizes developing quantile DP-fairness,
> > and constructing valid prediction intervals thereby, our CFQP approach bridge the gap in this respect.\
> > \
> > Methodologically, we incorporate a novel kernel smoothing step in the barycenter construction,
> > which is particularly advantageous for multiple subgroups of relatively small group sizes, the kernel smoothing method would provide more reliable quantile function estimations by reducing the random
> > variation and improving relative efficiency.\
> > \
> > Furthermore, we propose a conformalized fair quantile prediction interval (CFQP) method. It achieves a finite sample, distribution-free validity while satisfying demographic parity on different quantiles. We also provide the theoretical justifications for
> > the CFQP, including the exact DP guarantee of the fair quantile estimators as well as the valid control of prediction interval coverage.
> >
> > References:\
> > [1] A. Agarwal, M. Dudík, and Z. S. Wu. Fair regression: Quantitative definitions and reduction-based
> > algorithms. In International Conference on Machine Learning, page 120–129. PMLR, 2019.
> >
> > [3] M. J. Budig and M. J. Hodges. Statistical models and empirical evidence for differences in the motherhood
> > penalty across the earnings distribution. American Sociological Review, 79(2):358–364, 2014. doi:
> > 10.1177/0003122414523616.
> >
> > [7] J. García, P. J. Hernández, and A. López-Nicolás. How wide is the gap? an investigation of gender
> > wage differences using quantile regression. Empirical Economics, 26(1):149–167, Mar 2001. doi:
> > 10.1007/s001810000050.
> >
> > [8] J. Gardeazabal and A. Ugidos. Gender wage discrimination at quantiles. Journal of Population Economics,
> > 18(1):165–179, Mar 2005. doi: 10.1007/s00148-003-0172-z.
> >
> > [11] A. Killewald and J. Bearak. Is the motherhood penalty larger for low-wage women? a comment on quantile
> > regression. American sociological review, 79(2):350–357, 2014.
> >
> > [12] E. Chzhen, C. Denis, M. Hebiri, L. Oneto, and M. Pontil. Fair regression with wasserstein barycenters.
> > Advances in Neural Information Processing Systems, 33:7321–7331, 2020.
> >
> > [15] R. Williamson and A. Menon. Fairness risk measures. In International Conference on Machine Learning,
> > pages 6786–6797. PMLR, 2019.
> >
> > [42] Y. Romano, E. Patterson, and E. Candes. Conformalized quantile regression. In Advances in Neural
> > Information Processing Systems, volume 32. Curran Associates, Inc., 2019.

---

> ### Comment · Reviewer_3BdP · 2022-08-07
> **After the response**
>
> Thank you for a thorough response. It adequately addresses all of my queries. The presentation of the problem statement and overview of the method has been greatly improved in the newer submission. I appreciate the honest response to comparison with related work [1] and meticulous comparison in running time.
>
> The work addresses an important problem via a principled approach. Thus, moving the fair learning field forward beyond notions based on averages to quantiles. Hence I have increased my score to 7, Accept.

---

> > ### Author Response · Authors · 2022-08-08
> > **Reply to reviewer 3BdP.**
> >
> > Thank you so much for reconsidering our paper. We really appreciate your detailed and constructive suggestions and comments which help improve the paper greatly.

---

### Meta-Review · Area_Chair_5CV7 · 2022-08-26

**Recommendation:** Accept
**Confidence:** Less certain

**Metareview:**

This work extends the group-level fairness definitions (that were primarily established for supervised learning tasks) to the problem of conformalized quantile regression. A conceptual contribution is to redefine the group-level fairness using the average prediction to quantiles. Based on this adaptation, the authors further developed a postprocessing technique to revise a trained quantile regressor to satisfy the modified fairness definition for quantiles.

All reviewers acknowledged that the paper is reasonably written and the main idea delivers smoothly. A mixture of theoretical and experimental results are provided

There were some questions raised prior to the rebuttal and were successfully addressed, including clarifying the running time of computing quantiles, comparing them to other fair approaches, and partially misinterpreting DP. The authors are strongly encouraged to incorporate these comments into the final version.

Some reviewers had remaining questions about the novelty of the paper in light of prior results but the meta reviewer feels the introduced concept of fairness on quantile can be a good addition to the literature and might inspire follow-up works.

**Award:**

No

---

### Decision · Program_Chairs · 2022-09-14

Accept